# ATTENTIVE CROSS-MODAL PARATOPE PREDICTION

**Andreea Deac & Petar Veličković**
Department of Computer Science and Technology
University of Cambridge
Cambridge, CB3 0FD, UK
`{aid25,pv273}@cam.ac.uk`

**Pietro Sormanni**
Department of Chemistry
University of Cambridge
Cambridge, CB2 1EW, UK
`{ps589}@cam.ac.uk`

## ABSTRACT

Antibodies are a critical part of the immune system, having the function of directly neutralising or tagging undesirable objects (the antigens) for future destruction. Being able to predict which amino acids belong to the paratope, the region on the antibody which binds to the antigen, can facilitate antibody design and contribute to the development of personalised medicine. The suitability of deep neural networks has recently been confirmed for this task, with Parapred outperforming all prior physical models. Our contribution is twofold: first, using just the antibody data, we outperform the results of Parapred by producing a model which is computationally significantly more efficient by using à trous convolutions and self-attention. Secondly we implement cross-modal attention by allowing the antibody residues to attend over antigen residues. This leads to new state-of-the-art results on this task, along with insightful interpretations.

## 1 INTRODUCTION

Antibodies are Y-shaped proteins used by the immune system to neutralise pathogens such as bacteria and viruses. This is done when the antibody binds to the unique molecules on the pathogen called antigens. With antibodies being the most important class of biopharmaceuticals, knowing which amino acids are needed for the binding is a type of information that can have a significant impact on applications in diagnostics and therapeutics. In particular, creating novel antibodies requires the optimisation of properties such as solubility and stability, for which the non-binding amino acids can be used while maintaining the same binding affinity.

Traditional attempts for predicting the binding amino acids (the *paratope*) were based on hard coded physical models, requiring vast amount of information. Predictors such as Antibody i-Patch (Krawczyk et al., 2013) use as input the antibody's and the antigen's structure, while proABC (Olimpieri et al., 2013) needs the entire antibody sequence and additional features including the antigen volume.

Only recently Parapred (Liberis et al., 2017), a hybrid architecture consisting of convolutional and recurrent layers has become the state of the art technique. However, its usage of recurrent layers represents a significant performance bottleneck, and it discards the information about the target antigen entirely.

In this work, we ouperform Parapred by addressing its limitations and leveraging the bleeding-edge techniques in the language modelling community such as à trous convolutions (Kalchbrenner et al., 2016) and self-attention (Vaswani et al., 2017), while also significantly lowering computation time. We then manage to further improve this result by *cross-modally attending* over sequential antigen information, managing to derive qualitative insights from the attentional coefficients in the process.

## 2 DATASET AND PREPROCESSING

We used a subset of the Structural Antibody Database (SAbDab) (Dunbar et al., 2014), which provides antibody-antigen complexes, in order to train and test our models. The subset and extracted features were chosen to be the same as in Parapred. In addition, a one-hot encoding of the chain the residue belongs to was added to its features (thus each residue having 34 features) and the antigen

residues were extracted similarly for the Antibody-Antigen method. The final dataset comprises 239 antibody-antigen complexes.

## 3 METHODS

### 3.1 ANTIBODY-ONLY

We build up on the developments of Parapred by substituting recurrent layers with a combination of à trous convolutional layers (for efficient modelling of longer-range dependencies) and a self-attentional layer (allowing for efficiently covering the entire sequence). We will refer to this architecture as *Fast-Parapred* for the remainder of this paper.

The architecture consists firstly of a stack of four à trous convolutional layers, each with kernel size 3 and dilation rate 1,2,4 and 8 respectively. Then self-attention is applied on the computed features, leveraging the same attention mechanism as utilised by Veličković et al. (2017). Lastly a pointwise convolutional layer and the logistic sigmoid non-linearity are applied, in order to classify each considered antibody amino acid as binding or non-binding.

The regularisation methods used are dropout (Srivastava et al., 2014) (with $p = 0.5$ on the final layer and $p = 0.15$ on all the other ones), batch normalisation (Ioffe & Szegedy, 2015) on the output of each layer, and a skip connection (He et al., 2015) over self-attention.

### 3.2 ANTIBODY-ANTIGEN

With similar motivation as before, we extract features from antibody and antigen amino acid residues by applying, independently to both, a stack of four à trous convolutional layers (with exactly the same hyperparameters as for the antibody-only model). The self-attention in the antibody-only paratope predictor is then replaced with cross-modal attention over the antigen residue features. We will refer to this architecture as *AG-Fast-Parapred* for the remainder of this document.

The input to our cross-modal attention layer is a set of antibody residue features $\mathbf{b} = \{\vec{b}_1, \vec{b}_2, ...\vec{b}_M\}$, a set of antigen residue features $\mathbf{g} = \{\vec{g}_1, \vec{g}_2, ...\vec{g}_N\}$ and for each antibody residue $\vec{b}_i$ a set $\nu_i$ which marks the antigen residues which are in a fixed-range neighbourhood from $\vec{b}_i$. This neighbourhood was chosen to restrict the number of antigen residues being attended over by any antibody residue to 150. In addition we apply weight matrices $\mathbf{W}_1$ and $\mathbf{W}_2$ which represent learned linear transformations applied to $\mathbf{b}$ and $\mathbf{g}$, respectively. The attentional coefficients are then computed using the attention mechanism $a$ (as used by Veličković et al. (2017)), using the antibody residues as the *queries* and antigen features as *keys and values*. These coefficients are subsequently normalised using a softmax activation:

$$\alpha_{ij} = \frac{\exp\left(a\left(\mathbf{W}_1\vec{b}_i, \mathbf{W}_2\vec{g}_j\right)\right)}{\sum_{k\in\nu_i}\exp\left(a\left(\mathbf{W}_1\vec{b}_i, \mathbf{W}_2\vec{g}_k\right)\right)} \tag{1}$$

Using the normalised attention coeffcients, we then compute a linear combination of the corresponding antigen residues, for each attending antibody:

$$\vec{b}_i' = \sigma\left(\sum_{j\in\nu_i}\alpha_{ij}\mathbf{W}_2\vec{g}_j\right) \tag{2}$$

The result is, in a similar way to the Antibody-only method, passed through a pointwise convolutional layer and a logistic sigmoid non-linearity is applied, in order to classify each considered antibody amino acid residue as binding or non-binding.

We apply the same regularisation as for the antibody-only model—along with a skip connection over the cross-modal attention (allowing the antibody features to be combined with the obtained antigen features).

# 4 RESULTS

## 4.1 QUANTITATIVE RESULTS

We perform 10-fold crossvalidation on Parapred, Fast-Parapred and AG-Fast-Parapred. For each, we monitor ROC-AUC (which we also report for *proABC*; Table 1) and the precision/recall curve (which we also report for *Antibody i-Patch*; Figure 1), along with *95% confidence intervals*. Our results successfully demonstrate significant outperformance of *AG-Fast-Parapred*, for the first time successfully leveraging antigen information in a deep paratope predictor, while simultaneously relying solely on convolutional and attentional layers, removing the dependency on recurrent layers entirely.

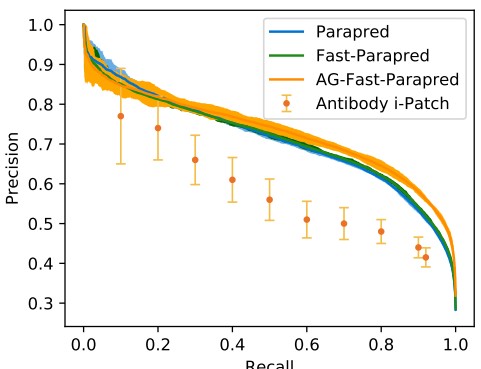

| Method | ROC AUC |
|---|---|
| **proABC** | 0.851 |
| **Parapred** | $0.880 \pm 0.002$ |
| **Fast-Parapred** | $0.883 \pm 0.001$ |
| **AG-Fast-Parapred** | $\mathbf{0.899 \pm 0.004}$ |

Table 1: Comparative evaluation results after 10-fold crossvalidation

Figure 1: Precision-Recall curves with highlighted 95% confidence intervals

## 4.2 QUALITATIVE RESULTS

We visualise, using PyMOL, the computed binding probabilities of *AG-Fast-Parapred*, on a test antibody-antigen complex, in Figure 2—revealing that its neural network has learnt to appropriately infer positional information (predicting higher probabilities for the residues closer to the antigen), *without being given any 3D coordinates*. The attentional coefficients computed by AG-Fast-Parapred are also visualised, for a single antibody residue, in Figure 3. From these we may observe that the attentional mechanism will tend to assign larger importances to antigen residues that are *closer* to this antibody residue—indicating the usefulness of the cross-modal attentional mechanism, and potentially hinting at a joint method for predicting antigen binding sites (*epitopes*), which we leave to future work.

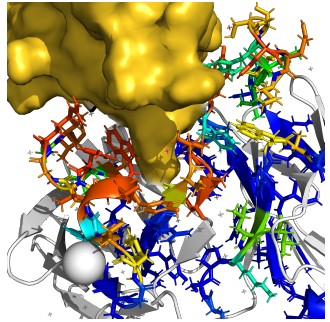

Figure 2: Antibody residue binding probabilities to the antigen (in gold). Warmer is higher.

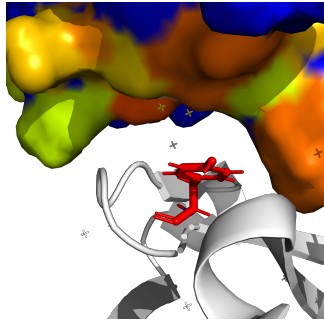

Figure 3: Normalised antigen attention weights for a single antibody residue (in red). Warmer is higher.

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
