# OpenReview forum: "Attentive cross-modal paratope prediction"
_ICLR.cc/2018/Workshop — Reject_

### Official Review · AnonReviewer1 · 2018-03-09
**Nice work, quite some missing details**

**Rating:** 5
**Confidence:** 4

**Review:**

The authors propose a convolutional architecture to predict binding amino acids from antibodies and (in one version) antigens and improve over previous state-of-the-art on a subset of the SAbDab database.

The clarity of the submission could be significantly improved. Especially since the setting of the paper is probably very unfamiliar for most researchers at ICLR I would have prefered to have not so much important information referred to the literature. For example: how does the data look, how is it fed into the convolutional architecture, how does the attention mechanism work, some more comparison to previous methods to give an impression of the magnitude of the performance increase. Especially the model architecture is far from clear.

The novelty of the paper seems to be mostly the cross-modal attention layer of the antibody-antigen model. Most other parts of the model seem to be taken from the literature and especially the attention mechanism of velickovic has already been used to model protein-interactions, but of course combining existing methods in a new way that sets a new state-of-the-art is also a valid contribution.

Except for the clarity issues pointed out above, I think think this could be a valuable contribution for the ICLR workshops. It shows a very relevant application of deep networks that is far from the standard tasks that most researchers work on.

---

### Official Review · AnonReviewer4 · 2018-03-11
**Interesting architecture and application but needs additional experiments**

**Rating:** 4
**Confidence:** 3

**Review:**

This submission proposes a deep neural network for the problem of paratope prediction -- an important problem in drug design. While a previous DNN has been shown to provide improved accuracy for this task, this paper proposes the use of specific one-dimensional convolutions and self-attention to obtain computational efficiencies over the previous approach.

The authors show that the new architecture is more accurate than the previous approach ParaPred on a benchmark dataset. However there are several issues with the current paper:

1. While the new architecture is motivated as being computationally more efficient, no evidence is provided of the time for learning and predicting using this architecture relative to ParaPred.

2. The gains in accuracy are modest. How were the standard errors computed ? If these are from 10-fold CV, then the predictions across folds cannot be treated as independent.

---

### Decision · Program_Chairs · 2018-03-20
**ICLR 2018 Workshop Acceptance Decision**

**Decision:**

Reject

**Comment:**

Based on the reviews, this paper has not been accepted for presentation at the ICLR workshop. However, the conversation and updates can continue to appear here on OpenReview.